# Spatial heterogeneity of internal migration in China: The role of economic, social and environmental characteristics

Haibin Xia[1], Liu Qingchun[2], Emerson Augusto Baptista[3]*

1 Key Laboratory of Geographic Information Science, Ministry of Education, School of Geographic Sciences, East China Normal University, Shanghai, China, 2 Institute of Regional Economy, Shandong University of Finance and Economics, Jinan, China, 3 El Colegio de México, Center for Demographic, Urban and Environmental Studies, Mexico City, Mexico

* ebaptista@colmex.mx

**Data Availability Statement:** The datasets generated and/or analyzed during the current study are publicly available at: http://www.stats.gov.cn/tjsj/ (in Chinese). Anyone is able to access these data in the same manner as the authors. The

## Abstract

The purpose of this paper is to explore the spatial heterogeneity of internal migration in China and to discuss the influence of economic, social and environmental characteristics on this demographic process. The overall results suggest that migration in China occurred from inland to coastal areas and from rural areas to urban areas. By stepwise regression, we identified that 9 out of 15 factors with potential influence on internal migration were retained, and the multicollinearity among them was reduced. In addition, we used the OLS and GWR regression analysis to discuss the global and local effects of relevant factors on internal migration. Economic scale (GDP), population concentration (population density) and demographic dividend (labour force proportion) were the three main driving forces of internal migration. In turn, internal migration further widened the gap of economic scale, population agglomeration and demographic dividend between counties and cities. Internal migration in southern coastal areas of China was most affected by economic aspects and demographic dividend. In the central China, the population was more concentrated in high-density cities, while in the eastern regions, areas with high level of education were conducive to immigration, thus forming talent reserve highlands. In the west, areas with highly educated level faced out-migration, which might cause brain drain and widen further the gap in talent reserves between the east and the west in China. From the perspective of location, the net immigration of the provincial capital was accompanied by the net immigration of the surrounding area, which was conducive to the formation of city clusters or urban sprawl. On the other side, the net immigration in prefecture-level cities often meant the net out-migration in surrounding areas. The correlation is particularly strong in eastern coastal provinces.

## Background

Since the beginning of the 21st century, China's population has been growing slowly but changing fast [1]. One of the main reasons for this rapid change is internal migration, which

authors did not have any special access privileges
that others would not have.

**Funding:** Specify the role(s) played the National
Social Science Fund of China(20BRK022)
Sponsored by Natural Science Foundation of
Shanghai (19ZR1415200).

**Competing interests:** The authors have declared
that no competing interests exist.

increased 81% between 2000 (fifth census) and 2010 (sixth census) Census [2]. The difference
in population growth rate between China's regions is gradually expanding, and can hardly be
explained by the difference in natural growth [3]. In other words, migration has replaced fertil-
ity and mortality as the main driver of Chinese population change [4].

For a long time, China seemed to worry about the growth of its population. The most typi-
cal example was the implementation of the family planning policy in 1978. However, with the
gradual decline in China's population growth rate, the country has experienced negative
growth in some areas. This situation has been cause for concern, because with the demo-
graphic dividend window decreasing, there will be an impact on economic growth in these
areas, while in others, such as coastal megacities, the government hopes to limit population
growth caused by immigration by a strict "hukou" [5] policy [6]. Therefore, more and more
attention has been paid to the study of China's population migration [7–11].

The discussion of migration started with analyze the spatial pattern and direction of popu-
lation flow [8,12–14]. In China, since the reform and opening up (1978), large numbers of peo-
ple were pouring into megacities such as Beijing and Shanghai [2]. The spatial change of
China's population showed a basic trend of agglomeration from rural to urban and from
inland to coastal areas [15]. Of course, different developmental stages of different countries or
regions also have different migration characteristics. For example, the population of Britain
has shown a trend of reverse flow from urban to rural areas, and the spatial agglomeration
degree of population has slightly decreased (Brown 2012). The flow of American immigrants
has fluctuated between urban and rural destinations in different regions [16]. Therefore, based
on discussing the number and direction of migration, people are more concerned about which
factors affect the potential and motivation of migration.

Some scholars have focused on the construction of logical framework of migration potential
and motivation. Cohen (1995) proposed five putative influences of migration, including demo-
graphic characteristics, socioeconomic conditions, transportation accessibility, natural ameni-
ties, and land development. Black et al. (2011) also proposed five drivers of migration,
including social drivers (education, family/kin, etc.), environmental (exposure to hazards, eco-
system services and others), economic (employment opportunities, income, etc.), demo-
graphic (population size, structure, etc.) and political (governance, conflict, etc.). Meanwhile,
in specific research, scholars tend to have different focuses. Economic factors, for example, are
the most widely discussed in migration studies. GDP per capita and unemployment rate seem
to be key determinants, and these changes drive migrants away from their areas and lead them
to "affluent" destinations [17]. The interprovincial migration in China since 1990s has been
largely driven by wage differentials [18]. Jennissen (2003) came to similar conclusions in
assessing the impact of economic factors on net migration in Western Europe from 1960 to
1998. Treyz et al. [19] found that net migration in the United States was induced by the
employment rate. With the continuous migration of population, population growth would
inhibit the employment rate and eventually stop net migration. Some scholars found that net
migration rate was an increasing function of the median family income or expected family
income and a decreasing function of the average cost of living [20,21]. In addition, transporta-
tion and urban infrastructure construction, which are closely related to regional economic
development, are also important factors to attract migrants [22–24].

In addition, demographic factors such as population density, education level, labor force
structure, and age structure are also considered as important migration factors. Molloy and
Smith (2011) discussed the impact of labor market friction on migration rate and found that
the reason why immigration rate of the United States was higher than that of Europe was that
the labor market friction of the former was lower. Since the 1990s, population of counties in
eastern Germany have been increasingly concentrated in cities such as Berlin, Leipzig and

Dresden, a trend driven by young people seeking better education and employment opportunities [25]. Pu et al. (2019) studied China's interprovincial population flows from 2005 to 2010 and found that population size was the most critical determinant, and the effect of spatial spillover on population size was significantly enhanced. Aronsson et al. (2001) found that the initial endowment of human capital (as measured by the proportion of the population with higher education) had a positive effect on the subsequent net migration in Sweden. Wang et al. (2019) found that unemployment and infant mortality in Russia were significantly negatively correlated with net migration, while urbanization rate, city size and life expectancy were significantly positively correlated, indicating that better job market, better economic status and health-related welfare were all factors that attract immigrants. Finally, Xiang et al. (2018), analyzing the spatial and demographic characteristics of the population flow of Wuhan, China, found that the migrant population consisted mainly of fertile women and young adults, while the elderly and children were left behind in the countryside.

In the late 20th century, due to concern about climate change, concepts such as climate migration and environmental migration came into the debate [26,27]. Early discussions of the environmental pressures of migration focused on the human-food relationship. Then, discussion included water, fossil fuels, timber, and other essentials [28]. Aronsson et al. (2001) found determinants of regional mobility are "fixed endowments" (related to geography and climate). Minale (2018) analyzed how rural households in China cope with negative productivity shocks in agriculture by rural-urban migration and off-farm work. The results showed that under the impact of 1 standard deviation of negative rainfall, agriculture decreased by 4.5% and migration increased by about 5%. Studies of rural (non-metropolitan) immigrants in the United States found that, consistent with studies of landscape preferences, the areas that people were most attracted to were forests and open land, waters, terrain variability, and relatively little arable land [29].

The following issues existed in previous research on internal migration. First, the analysis of potential influencing factors of internal migration often suffered from the contradiction between theoretical construction and practical application. Theoretical construction is expected to be comprehensive, while practical application needs to consider the constraints of data accessibility and data covariance. Second, the global regression results between internal migration and related potential factors, such as population density or education level, may have completely opposite findings on their relationship in the analyzed cases in different countries and regions, and thus their spatial heterogeneity must be investigated. Third, some of the studies tend to focus the impact of socioeconomic factors on internal migration and tend to discard the exploration of natural environmental factors, and vice versa. Therefore, the purpose of this paper is to explore the spatial heterogeneity of internal migration in China and to understand the impact of social, economic, and environmental factors on one of the demographic processes.

Usually, population migration is often not determined by a single factor. For this reason, multiple linear regression method is widely used in migration analysis [30]. However, the reasonable selection of influencing factors always seems to be a difficult problem. While it is necessary to absorb as many explanatory variables as possible, we should be aware that, as the number of included factors increase, the multicollinearity among factors might affect the final analysis results. In addition, we should pay more attention to that fact that migration both affects and is affected by economic conditions [31]. Finally, influencing factors tend to have varying degrees of impact in places. For example, migration driven by natural factors in Italy occurs only in certain areas, not at the national level [32]. Due to China's vast land area and large population, the degree of influence of potential influencing factors on migration varies spatially [33]. Compared with linear regression analysis, Geographically Weighted Regression

(GWR) model is usually used to explore the spatially varying relationships between dependent variables and potential influencing factors. The GWR model assumes that the regression coefficient changes as a locational function and focuses on the problem of explaining relationships in non-static spatial conditions that cannot be solved by the global model. The GWR model is a local spatial statistical technique for exploring spatial non-stationarity [34] and is widely used in the statistical spatial analyses of population distribution [35], mortality [36], and migration [37]. However, GWR is unable to perform the task of selecting the variables, and it is thus problematic when there are many multicollinear explanatory variables that cannot be ignored [38]. Several papers have been devoted to the multicollinearity of GWR [39,40]. Therefore, it is necessary to properly handle multicollinearity among factors and use OLS to analyse the global correlation of factors before conducting GWR analysis.

The purpose of this paper is to explore the spatial heterogeneity of internal migration in China and to understand the influence that social, economic and environmental factors have on this demographic process. The structure of this paper is as follows. Firstly, we will use the relevant data of China's fifth (2000) and sixth (2010) censuses to calculate internal migration in China in 2000 and 2010. Spatial characteristics of internal migration in 2000 and 2010 will be analysed. Secondly, we will select 15 factors potentially affecting internal migration from five aspects: economy, demography, environment, location and transportation, and land use. By stepwise regression analysis, the factors that have a significant impact on internal migration will be retained, and the multicollinearity among the retained factors will be minimized. Global regression results (OLS analysis) and local regression results (GWR analysis) of retained factors with internal migration will be analysed. The influence degree and distribution characteristics of related influencing factors on internal migration will be discussed. Finally, some policy suggestions on internal migration in China are given based on the analysis results.

## Data and methods

### Data

This paper will use the data of the fifth and sixth censuses of China to calculate net migration in Chinese counties and cities as proxy for internal migration. Net migration is given subtracting immigration by out-migration. There are no statistics on out-migration in the census data. Therefore, the difference in the size of the permanent population and the size of the registered population can be used to study the features and the scale of net migration [41].

The independent variables adopted in this paper are divided into five categories (Table 1). The first is economic factors, including GDP, GDP per capita, and income per capita of farmers. GDP represents the size of the market and economic scale. GDP per capita, roughly, represents economic development level and standard of living of residents. Income per capita of farmers reflects the income level of rural residents. This data comes from Statistical yearbook of counties and cities in China (2009–2011).

The second category is related to demographic factors, such as population density, proportion of labour force and years of education. Population density in China is most concentrated in the east of the country. The second one represents the proportion of workforce aged between 16 and 65 in the total population. A higher proportion of labour force indicates a larger potential employment population and higher level of demographic dividend. Years of education represents the level of education and intellectual reserve in different regions. Data of demographical factors were obtained from sixth population census of China (2010).

The third category (environmental) includes river density, accumulated temperature and soil nutrients. These three factors represent the potential for food production in different regions. Soil nutrients data is obtained from China Soil database (http://vdb3.soil.csdb.cn/).

**Table 1. Potential factors that influence internal migration and data description.**

| Category | Variables | Abbreviation | Database | Unit |
|---|---|---|---|---|
| **Dependent variable** | **Net migration** | **NetMig** | **China Census Data** | **thousand people** |
| Economic | GDP | GDP | Statistical yearbook | billion Chinese Yuan |
| | GDP per capita | GDPPerCapita | Statistical yearbook | thousand Chinese Yuan / people |
| | Income per capita of farmers | IncomeFarmer | Statistical yearbook | thousand Chinese Yuan |
| Demographic | Population density | PopDen | China Census Data | people / km$^2$ |
| | Proportion of the labour force | LaborForce | China Census Data | % |
| | Years of education | Edu | China Census Data | years |
| Environmental | River density | RiverDen | NGCC | km / km$^2$ |
| | Accumulated temperature | AccTemp | NGCC | ˚C |
| | Soil nutrients | Soil | China Soil database | |
| | Road density | RoadDen | OpenStreet | km / km2 |
| Traffic and Location | Distance to the nearest provincial capital | DistanceToCapital | ArcGIS | km |
| | Distance to the nearest prefecture-level city | DistanceToCity | ArcGIS | km |
| Land Use | Built-up proportion | BuiltUp | GHSL | % |
| | Cultivated land proportion | CropLand | SEDAC | % |
| | Forest cover proportion | ForestLand | SEDAC | % |

The other environmental data is from National Geomatics Center of China (NGCC) (http://www.ngcc.cn/).

The fourth category is traffic and location factors, including road density, distance to the nearest provincial capital, and distance to the nearest prefecture-level city. Road density represents infrastructure level and traffic accessibility, and the data is from OpenstreetMap (https://www.openstreetmap.org), while the two location factors reflect the distance from nearest central city. The data is calculated by ArcGIS software.

For land use factors, the variables built-up proportion, cultivated land and forest cover proportion were selected. Built-up proportion is the ratio of urban land use to the total area and reflects the level of land urbanization. Data source is from Global Human Settlement (GHSL) (https://ghslsys.jrc.ec.europa.eu/). The two other land use variables, cultivated land proportion and forest cover proportion, represent, respectively, the ratio of agricultural and forest land to the total area, and were obtained from SEDAC (http://sedac.ciesin.columbia.edu/). The data sources and dimensions of the independent variable are shown in Table 1.

## Methods

According to the definition of the Chinese census, the migrant refers to the population who are "in the household" at the time and place during the census, but whose hukou registration place has changed across townships within a specified period of time. The permanent population of counties and cities are the in-migrants and the local permanent population. On the other hand, registered population are the out-migrants and the local resident population. Therefore, immigrants ($I_i$), out-migrants ($O_i$), local permanent population ($L_i$), permanent population ($P_i$) and registered population ($H_i$) of county $i$ have the following relation:

$$L_i = P_i - I_i = H_i - O_i \tag{1}$$

Net migration (NetMig) is given by the difference between immigration and out-migration or the difference between the permanent and registered populations [42]. In this paper, we use

net migration of counties and cities to measure the level of internal migration.

$$NetMig_i = I_i - O_i = P_i - H_i \qquad (2)$$

After calculating net migration data in 2000 and 2010, we will conduct exploratory analysis of the relevant data. Firstly, net migration data in 2000 and 2010 will be drawn to analyse the basic spatial distribution characteristics. Secondly, LISA clustering tool will be used to analyse the spatial clustering characteristics of internal migration levels in China in 2000 and 2010.

The next step is to use stepwise regression to select a subset of "better" independent variables from among 15 variables that we have initially (see Table 1). The general idea behind the stepwise method is that we construct an "optimal" regression model from a set of candidate independent variables by inserting and removing predictors (one by one) into our model until there is no justifiable reason to enter or remove any more. This process is repeated until no insignificant independent variables are selected and no significant independent variables are removed from the regression equation. Adding or deleting a variable from the regression equation is a step along the way we make based on the partial *F*-tests that are obtained. In summary, stepwise method helps reduce multicollinearity.

After that, we use the ordinary least squares (OLS) regression, which is the starting point for development of a GWR model and one of the most popular statistical techniques used in the social sciences. Roughly, OLS expresses the relationship between the dependent variable and the explanatory variables, being possible also to identify the strength of the relationships between them.

In the OLS regression model, the dependent variable *y* (net migration) is statistically related to a set of *n* independent variables x as follows:

$$y_i = \beta_0 + \sum_{j=1}^{N} x_j \beta_j + \varepsilon_i \qquad (3)$$

where *i* represents each county or city; *m* denotes the number of independent variables; β$_0$ is the intercept; β$_j$ are the beta coefficients for each dependent variable obtained by stepwise regression; and ε is a randomly distributed error term. An OLS regression model can be converted into a GWR model by substituting each beta coefficient (the intercept and the dependent variable coefficients) with its local counterpart such that the beta-coefficients can vary across space:

$$y_i = \beta_{0(u_i,v_i)} + \sum_{j=1}^{N} x_j \beta_{j(u_i,v_i)} + \varepsilon_i \qquad (4)$$

where *i* represents each county or city and $(u_i, v_i)$ is the location in geographic space of the *ith* observation. A set of beta-coefficients (and hence a regression model) is estimated at each location based only on neighbouring geographically weighted data cells. A key feature is the ability to calibrate the spatial weighting function to identify the bandwidth, i.e., the number of or proximity of neighbouring cells included that results in a 'best-fit' model. In this paper, the AICc (Corrected Akaike Information Criterion) is used as the bandwidth method. Moran's I [43,44] is used to examine the spatial autocorrelation of the standardized residuals. If Moran's I is close to zero, the residual is randomly distributed and the model fits well.

We then perform a comparative analysis between OLS and GWR to better understand temporal and spatial changes of the Chinese migration. This analysis is important, since OLS regression assumes that the relationships between the variables are homogenous across space. However, spatial dependencies are often not homogenous across large geographical regions [45]. To address this, a GWR model is used to explore the spatially varying relationships between a dependent variable and potential influencing variables [35,46].

## Analysis

### Analysis of China's net migration

Fig 1 shows the distribution of net migration in China in 2000 and 2010 at the county and city levels. Net migration in China is characterized by the concentration of population mainly in the provincial capitals and some important prefecture cities. The red colour means net immigration and the blue colour means net out-migration. The Pearl River delta, the Yangtze River delta, and Beijing-Tianjin regions are the most obvious areas of immigration. On the other hand, the areas of out-migration are different in 2000 and 2010. In 2000, it was mainly concentrated inland areas of southern China, while in 2010, it was mainly concentrated in areas along the Yangtze River in central China. In the vast western regions, such as Xinxiang, Tibet, Inner Mongolia, Sichuan, Gansu and Yunnan provinces, net migration scale was relatively small, except for the capital cities. According Fig 1, and similar to the conclusion of Buhaug and Urdal [15], in general, China's population migration showed the trend from county to central city and from inland to coastal areas.

In order to further explore the spatial clustering characteristics of internal migration, we used Univariate Local Moran's I function of Geoda software to Chinese migration in 2000 and 2010 (Fig 2). Spatial adjacency was set to Queen adjacency and distance weight was set to Euclidean distance. High-High spatial clusters mean that both local and neighbours are all net immigration, while low-low spatial clusters means that both local and neighbours are in net out-migration. High-low spatial outliers implies that local is net immigration and neighbours is net out-migration, while low-high spatial outliers mean that local is net out-migration and neighbours is net immigration.

In 2000, high-high spatial clusters were mainly concentrated in the Pearl river and Yangtze river delta regions, while in 2010, the scope of high-high clustering in Pearl river delta and Yangtze river delta were further expanded, and the Beijing-Tianjin regions in the north also formed new cluster. Low-High spatial outliers, obviously, is located in the surrounding areas of high-high clusters. In high-high clusters, there was a spillover effect, causing people to move to surrounding areas or even further away. High-low spatial outliers was mainly distributed in

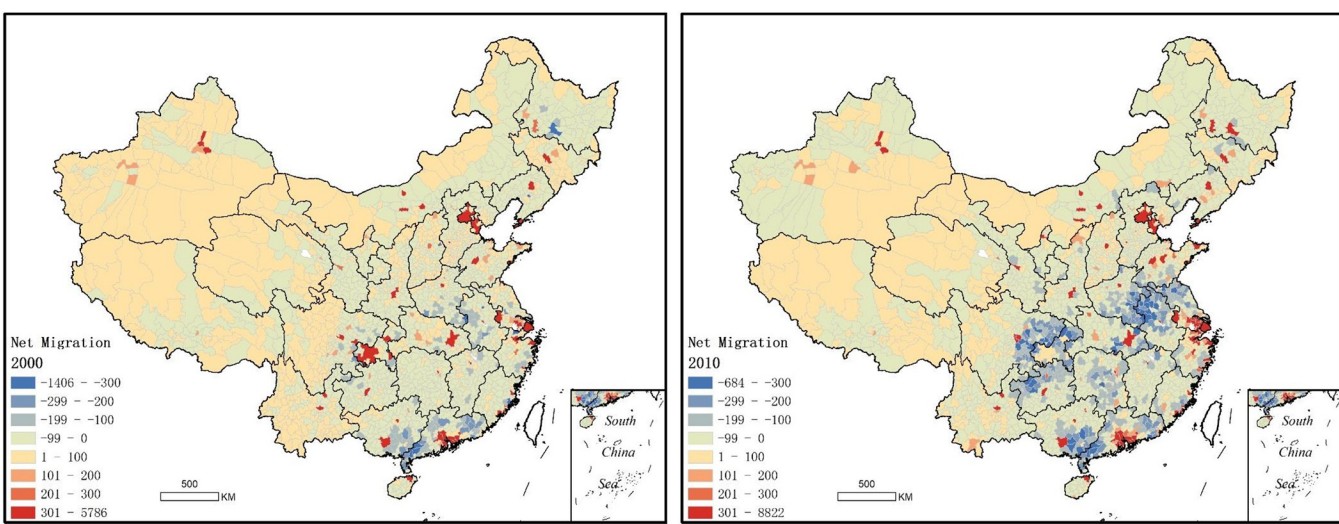

**Fig 1.** Net migration in 2000 (left) and 2010 (right).

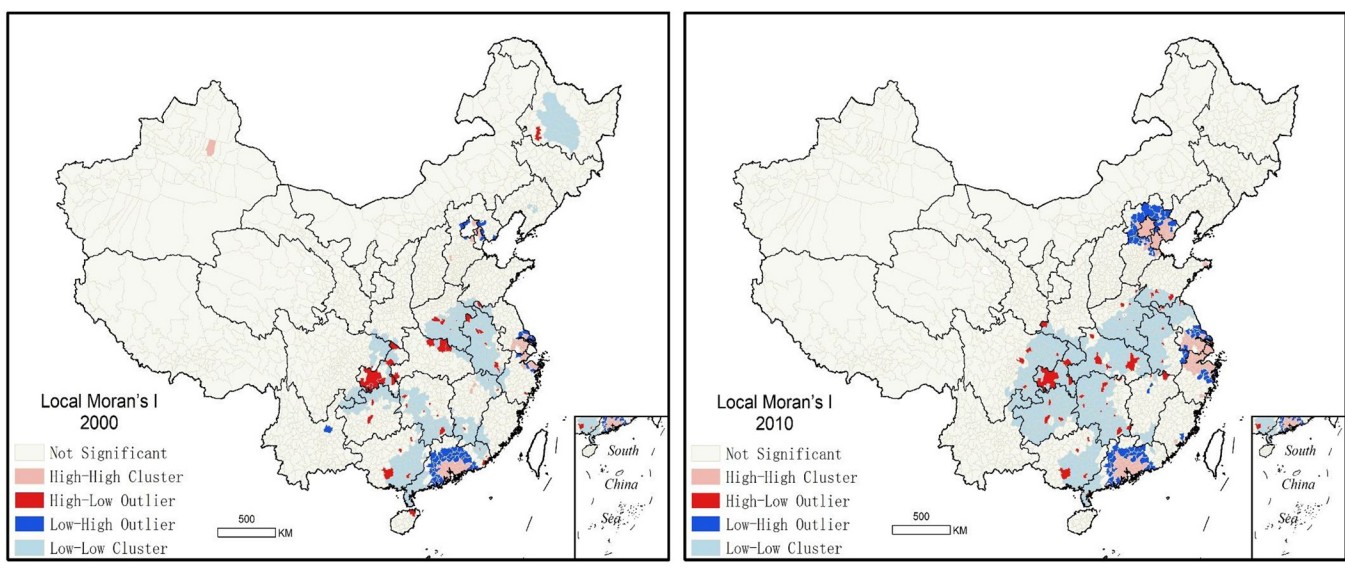

**Fig 2.** *LISA* cluster map of net migration in 2000 (left) and 2010 (right).

the capital cities and some prefecture-level cities in east and central China. The immigration itself means out-migration in surrounding area. In net migration clustering of China in 2000 and 2010 (Fig 2), the biggest difference is was in the distribution of low-low clustering. In 2000, low-low clustering was mainly distributed in parts areas of provinces in Pearl River Basin, and in the junction areas between provinces in Yangtze River Basin, where the economy development was relatively backward. Already in 2010, out-migration covered almost all areas of China's inland provinces along the Yangtze River, being expanded even further. In 2000, the out-migration was probably caused by economic backwardness, while in 2010, people probably were actively seeking higher living and working environment.

## Stepwise regression analysis

Table 2 presents the results of the stepwise regression method, where 9 variables are maintained from the 15 initial independent variables. It is them: GDP (economic); population density, years of education, and proportion of labour force (demographic); soil nutrients (environmental); distance to the nearest provincial capital and distance to the nearest prefecture-level city (traffic and location); and forestland and cultivated land proportion (land use). By stepwise regression, explanatory variables are significant for dependent variable, and multi-collinearity among them is effectively reduced. The highest VIF value is 2.571 (Table 3), which is well below the common cut-off point of 10 [47].

GDP per capita, income per capita of farmers, river density, accumulated temperature, road density and built-up proportion are excluded from stepwise regression model. Among all factors, one variable at least in each category was reserved. Of course, the excluded variables are only statistical results, which could also give some reasonable explanations. For example, GDP per capita is generally considered a pull force of migrants [17]. However, the neoclassical growth model reveals that population growth positively contributed to income per capita while the modified endogenous growth model showed a negative relationship between these two variables [48]. Whether the population growth of central cities derived the growth of

**Table 2. Stepwise regression analysis.**

| | Model 1 | Model2 | Model3 | Model 4 | Model 5 | Model 6 | Model 7 | Model 8 | Model 9 |
|---|---|---|---|---|---|---|---|---|---|
| GDP | 0.798** | 0.688*** | 0.694*** | 0.659** | 0.664*** | 0.667*** | 0.666*** | 0.670*** | 0.672*** |
| | (0.000) | (0.000) | (0.000) | (0.000) | (0.000) | (0.000) | (0.000) | (0.000) | (0.000) |
| Population density | | 0.169*** | 0.172*** | 0.179*** | 0.181*** | 0.180*** | 0.177*** | 0.177*** | 0.177*** |
| | | (0.000) | (0.000) | (0.000) | (0.000) | (0.000) | (0.000) | (0.000) | (0.000) |
| Crop land | | | -0.108*** | -0.110*** | -0.068 *** | -0.061*** | -0.069*** | -0.064*** | -0.048 *** |
| | | | (0.000) | (0.000) | (0.000) | (0.000) | (0.000) | (0.000) | (0.000) |
| Labor Force | | | | 0.100*** | 0.130*** | 0.132*** | -0.134*** | 0.155*** | 0.164 *** |
| | | | | (0.000) | (0.000) | (0.000) | (0.000) | (0.000) | (0.000) |
| Soil | | | | | -0.091*** | -0.082*** | -0.083*** | -0.078*** | -0.073*** |
| | | | | | (0.000) | (0.000) | (0.000) | (0.000) | (0.000) |
| Distance to city | | | | | | 0.044*** | 0.090*** | 0.092*** | 0.104*** |
| | | | | | | (0.000) | (0.000) | (0.000) | (0.000) |
| Distance to capital | | | | | | | -0.064*** | -0.067*** | -0.071*** |
| | | | | | | | (0.000) | (0.000) | (0.000) |
| Education | | | | | | | | -0.041*** | -0.050*** |
| | | | | | | | | (0.000) | (0.000) |
| Forest Land | | | | | | | | | 0.032*** |
| | | | | | | | | | (0.000) |
| $R^2$ | 0.637 | 0.654 | 0.665 | 0.674 | 0.680 | 0.681 | 0.683 | 0.684 | 0.685 |
| Adjusted $R^2$ | 0.637 | 0.653 | 0.665 | 0.674 | 0.679 | 0.680 | 0.682 | 0.683 | 0.683 |

Dependent variable: Net migration 2010.

Signif.codes

*p≤0.05

**p≤0.01

***p≤0.001.

**Table 3. Coefficients of OLS regression analysis and GWR analysis.**

| | Coefficients of Global model (OLS) | | | | Spatial Coefficients of Local model (GWR) | | | | | |
|---|---|---|---|---|---|---|---|---|---|---|
| | Coefficient | Std. Coefficient | p>t | VIF[1] | Minimum | Lower quartile | Median | Upper quartile | Maximum | StdDev |
| GDP | 0.300 | 0.672 | 0.000 | 1.908 | 0.130 | 0.197 | 0.266 | 0.301 | 0.459 | 0.077 |
| Population density | 0.021 | 0.177 | 0.000 | 1.743 | -0.001 | 0.016 | 0.023 | 0.037 | 0.061 | 0.014 |
| Crop land | -0.831 | -0.048 | 0.002 | 1.779 | -2.293 | -1.498 | -0.538 | 0.255 | 0.849 | 0.949 |
| Labor Force | 13.699 | 0.164 | 0.000 | 1.862 | -0.674 | 9.109 | 12.701 | 15.206 | 20.762 | 4.989 |
| Soil | -231.766 | -0.073 | 0.000 | 1.679 | -428.143 | -259.579 | -204.48 | -167.627 | -7.729 | 88.859 |
| Distance to city | 0.369 | 0.104 | 0.000 | 2.571 | 0.202 | 0.589 | 1.145 | 1.651 | 2.357 | 0.617 |
| Distance to capital | -0.200 | -0.071 | 0.000 | 2.407 | -0.575 | -0.329 | -0.241 | -0.169 | -0.091 | 0.124 |
| Education | -17.033 | -0.050 | 0.002 | 1.805 | -22.646 | -14.353 | 10.771 | 36.881 | 59.138 | 25.680 |
| Forest Land | 0.611 | 0.032 | 0.031 | 1.566 | -0.409 | 0.079 | 0.514 | 1.049 | 2.159 | 0.646 |
| Constant | -808.24 | | 0.000 | | -1,676.89 | -1,227.97 | -877.15 | -694.8 | -174.59 | 362.69 |
| Adjusted $R^2$ | 0.683 | | | | 0.799 | | | | | |
| AICc | 31,258 | | | | 30,221 | | | | | |

[1]Variance inflation factor (VIF) = measure of multicollinearity among the independent variables.

surrounding areas or attracted the population of surrounding areas might vary from region to region and stage of development [49]. Higher income per capita of farmers might mean that people are willing to stay on the farm, but it might also allow them to bear the costs of migration for better individual development. As a result, the relationship between GDP per capita / income per capita of farmer and internal migration might be uncertain on a global scale. River density and accumulated temperature not only affect agricultural production, but also industrial production and human life, so how will it ultimately affect internal migration might be uncertain.

In addition, there is significant multicollinearity among multiple independent variables. This means that after the variable that is the highest correlation with the dependent variable is selected in the stepwise regression process, other independent variables with multicollinearity will not be selected. For example, the size of the economy is usually linked to level of infrastructure construction, such as transportation and urbanization level. As GDP was included in the stepwise regression, road density and built-up proportion are removed. However, this does not mean that the level of infrastructure construction represented by road density and built-up proportion has no influence on internal migration, but this influence was indirectly reflected through GDP.

In summary, by stepwise regression, we extract significant variables from the various categories potentially affecting internal migration and minimize their multicollinearity. This also make a foundation for the subsequent spatial weight regression analysis.

## OLS regression analysis

Before the GWR analysis, we first analyse the correlation between 9 independent variables and net migration by the OLS method (Table 3). OLS analysis allows us to explore the relationship between net migration and potential influencing factors on a global scale, and the residual sum of squares is minimized for estimation. In addition, to examining the influence of independent variables on dependent variables globally, people tend to pay more attention to which factors have more obvious influence on internal migration locally.

Variables that are positively correlated with internal migration are: GDP, population density, proportion of labour force, distance to the nearest prefecture-level city, and forest cover proportion. The correlation coefficient of GDP is as high as 0.672, which means that GDP is the main driver of migration. Population density and proportion of labour force, which represent demographic aspects, also show positive correlation with net migration. The standardized correlation coefficient of population density is 0.177, second only to the standardized correlation coefficient of GDP. Demographers had long discussed population size as a manifestation of pressure on a carrying capacity [50]. Spatially, the agglomeration benefits formed by high population density attract more migrants to move in, while areas with low population density face the problem of out-migration. Meanwhile, the standardized correlation coefficient of the proportion of labour force is 0.164. A higher proportion of labour force means a higher demographic dividend. This shows that the bigger demographic dividend, the more ability to attract immigrants. Distance to the nearest prefecture-level city shows a standardized correlation coefficient of 0.104, which means people tend to move out of areas closer to prefecture-level cities. Finally, forest cover proportion has a significant positive correlation with net migration, with a standardized correlation coefficient of 0.032. People seem to be more willing to move to areas with higher forest cover.

Variables negatively correlated with population migration include years of education, soil nutrients, distance to the nearest provincial capital, and cultivated land proportion. The standard correlation coefficient of years of education is -0.050. The areas with higher education

level are more conducive to out-migration on a global scale. Soil nutrients and proportion of cultivated land area are closely related to agricultural development, and their standard correlation coefficients with net migration are -0.073, and -0.048, respectively. In the areas with good agricultural conditions, more people are willing to move out. Lastly, distance to the nearest provincial capital presented a standardized correlation coefficient of -0.071. Areas closer to provincial capitals seem to be more conducive to immigration.

## GWR analysis

GWR analysis can not only explore the impact of potential influencing factors on spatial heterogeneity of internal migration, but also provide better fitting results than OLS. The Gauss function method is chosen for spatial weight, because the Gauss function method can choose a continuous monotonically decreasing function to represent the relationship between weight and distance, which overcomes the shortcoming of the inverse distance method assuming stronger correlation for close features. Bandwidth value is based on AICc value to automatically find the optimal actual distance or adjacent elements. Table 3 shows the comparison of relevant parameters between the OLS and GWR methods. Adjusted $R^2$ of GWR analysis was 0.799, larger than OLS's 0.683; AICc was 30,221 (lower than OLS's 31,258), which indicates that the GWR had an improved model fit (Table 3).

Fig 3 shows spatial correlation coefficient distribution of the nine variables retained by the stepwise regression method, where the colors in shades of red represents the positive spatial correlation between the independent variables and internal migration; colors in shades of blue represents the negative correlation between them; and the gray part (hatches) represents the spatial correlation coefficient which could not pass the hypothesis test. This paper defined as 95% confidence interval. No data for Hong Kong, Macao and Taiwan, shown in white.

In OLS analysis, the relative contribution rate of GDP to net migration is the highest, with standardized coefficient of 0.672 (Table 3). The spatial correlation coefficient (GWR) shows positive correlation from the minimum value 0.130 to the maximum value 0.459, and the median is 0.266 (Table 3), gradually decreasing from the south to the central areas of China and gradually increasing from the central areas to the north (Fig 3A). Migrants in China's southern coastal provinces and areas around the capital (Beijing) are more influenced by economic scale.

Population density represents the degree of population agglomeration. The standardized coefficient is 0.177, just behind economic factor GDP, indicating that places with higher population agglomeration are more likely to attract immigrants. The spatial correlation coefficient between population density and internal migration is positive in almost all of China, except for a small part of northern areas that do not pass the hypothesis test (Fig 3B). The highest spatial correlation is observed in the provinces along the Yangtze River in central China, where the country' highest population densities are found. The Yangtze River basin account for three of China's six city clusters, including the urban agglomerations of the Yangtze River Delta, Wuhan and Chengdu-Chongqing. The development of these urban agglomerations absorbed a large number of migrants, mainly in the surrounding agricultural areas. On the other side, the reason that the spatial correlation between population density and migration is not significant in Beijing area might be related to its strict household registration system. Beijing's high population density has not led to a large number of migrants.

The proportion of labour force can be regarded as the level of demographic dividend, whose relative contribution rate to population migration is 0.164. In the GWR regression (Table 3), the spatial coefficient is between -0.674 and 20.762. The areas with higher demographic dividend usually have the greater potential for economic development, which, in turn, attracts more migrants. As a result, areas that previously had a demographic dividend

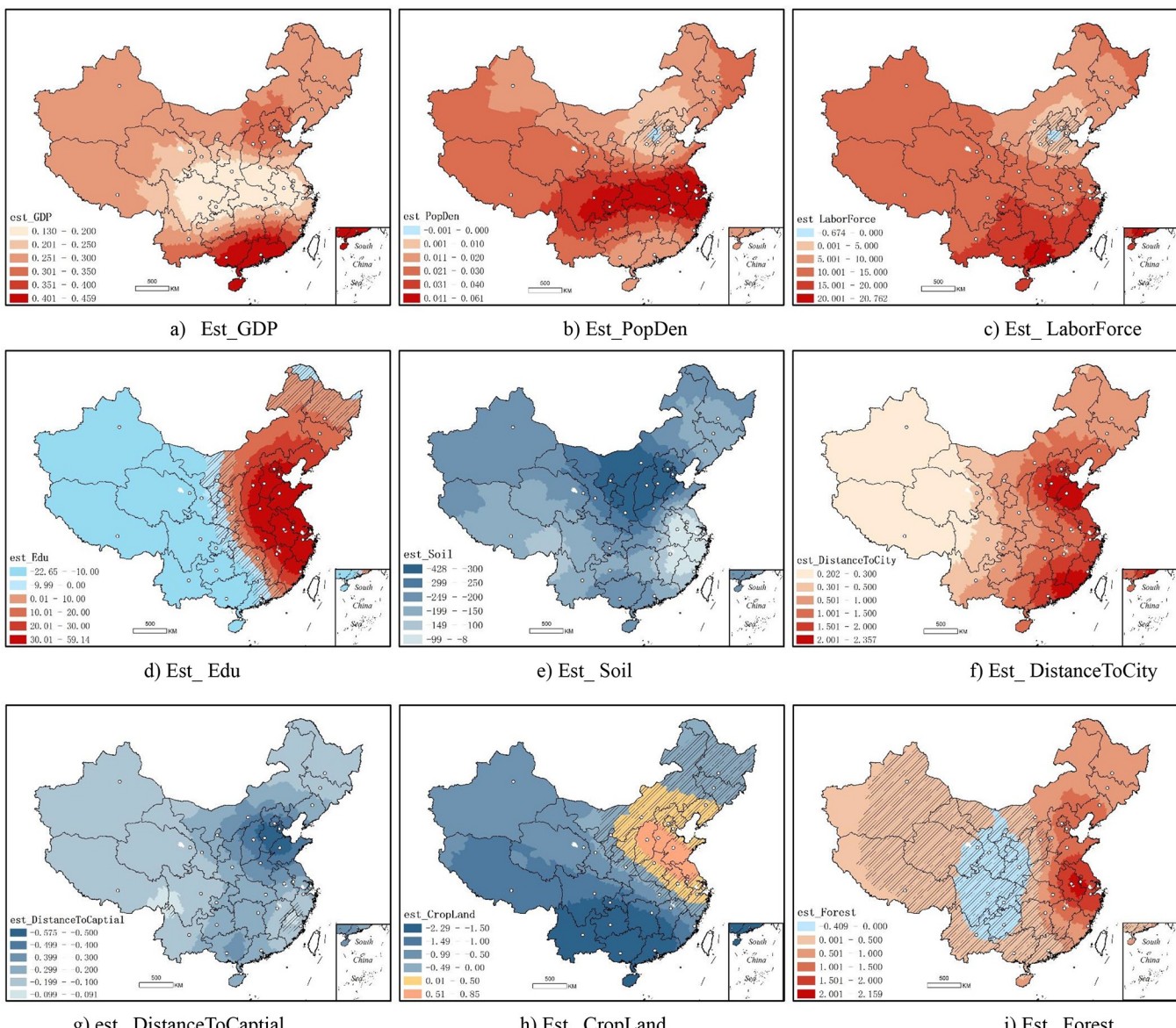

**Fig 3. Spatial correlation coefficient distribution of the impact factors.** a) Est_GDP, b) Est_PopDen, c) Est_ LaborForce, d) Est_Edu, e) Est_Soil, f) Est_DistanceToCity, g) est_DistanceToCaptial, h) Est_CropLand, i) Est_ Forest.

advantage are likely to maintain this advantage by internal migration. The spatial correlation between labour force proportion and internal migration shows a trend of decreasing from south to north (Fig 3C). Southern China's economy was growing fast and was more dependent on migrants since the reform and opening up (1978). For some of China's southern coastal migration cities (such as Shenzhen), a large number of migrant workers make the demographic dividend maintained at a high level.

Regarding education, there is a negative correlation with net migration. The standardized correlation coefficient is -0.05. Well-educated individuals have more migration opportunities than unskilled individuals [51]. Therefore, areas with higher levels of education are more likely to experience out-migration. From the perspective of spatial correlation coefficient distribution,

the values vary from a minimum of 0.130 to a maximum of 0.459, with a median of 0.266 (Table 3). Years of education in eastern China are positively correlated with internal migration, and the correlation has gradually decreased from coastal areas to inland areas. In the vast western areas, the two were negatively correlated (Fig 3D). This indicates that in the eastern areas, especially in the coastal zone, areas with a higher education level are more likely to attract migrants. The opposite happens in the western areas. Economically developed areas in the east tend to be highly educated, becoming ideal places for migrants to go. The highly educated population of the western might migrate to the eastern region in search of better job opportunities and quality of life, which leads to a widening gap in the education reserve between regions.

Soil nutrients are negatively correlated with internal migration. The standardized correlation coefficient is -0.073. The spatial correlation coefficient ranged from -428.143 to -7.729 (Table 3). An area with better agricultural conditions often means a larger number of rural residents. In the spatial coefficient distribution diagram (Fig 3E), the value of negative correlation is stronger in the inland areas of northern China. Under the general trend of internal migration from inland to coastal areas, farmers in inland areas became the main force of out-migration. In addition, people in areas with better agricultural production conditions might be able to afford the cost of migration and became the main source of out-migration.

The distances to nearest provincial capital and prefecture-level city are negatively and positively correlated with net migration, respectively. The spatial correlation coefficient of the first ranged from -0.575 to -0.091, while the second ranged between 0.202 and 2.357. The results suggest that areas closer to the provincial capital are more likely to be where of net immigration. On the other hand, areas closer to prefecture-level cities in China tend to show net out-migration. This trend can be seen in Fig 3F and 3G, where a decrease is noted from the eastern coast to the western inland. The development of provincial capitals (Fig 3G) in eastern coastal provinces has at the same time brought about a development of surrounding areas, which, together, become the areas with net immigration. The immigration of prefecture-level cities (Fig 3F) often means the out-migration of surrounding areas, especially in the coastal areas of north China. The population growth of mega cities, such as provincial capitals, plays a leading role in the population growth of surrounding areas. Meanwhile, the population growth of medium-sized cities, such as prefecture-level cities, is often accompanied by out-migration of surrounding areas. The former shows the spatial spillover relation of immigrants, while the latter shows the spatial competition relation.

Finally, the standardized correlation coefficient between the proportion of cultivated land and internal migration is -0.048. The GWR coefficient ranges from -2.293 to 0.849 (Table 3). The negative correlation degree of spatial coefficient gradually decreases from southwest to northeast (Fig 3H). This shows that urban land use and agricultural land use have a certain relationship with population migration, that is, the population gradually moved out of agricultural land areas to into urban areas, which is more significant in the southwest.

This paper reached a conclusion similar to McGranahan's (2008) about his study in the United States. In other words, consistent with studies of landscape preferences, people leave areas of cultivated land, being attracted to forests regions and open land. In this sense, eastern seaboard, where the areas have greater forest cover, is ideal for immigration. In fact, with the migration from rural to urban areas, the skills employed in rural areas are often no longer needed and people begin to look for alternatives for a better life and work environment.

## Discussion

The main characteristics of population migration in China are from rural to urban areas; and from inland to coastal areas. Migration had a bigger impact on population change in 2010

than in 2000. This conclusion follows the findings of Bell et al. [4], where migration played a decisive role in the redistribution of the Chinese population.

Economic agglomeration, population agglomeration and age-appropriate labour agglomeration are often complementary and tend to create scale effects, i.e., population prefers to move to places with high economic and population concentrations. There is a vast literature discussing the role of GDP in migration [4,23,52]. The areas with larger economy scale often mean better infrastructure, more potential jobs and so on, making them the primary choice for people to move in. From the perspective of China's current development, the agglomeration advantage brought by high population density (such as urban areas) is higher than the pressure brought by population crowding, which promote more people to migrate to areas with higher population density. Areas with a demographic dividend advantage are more likely to attract immigrants. The areas with higher level of GDP, population density and labour force proportion means more people to move in, and thus form the Matthew effect in space (the stronger gets stronger and the weaker get weaker). The areas with these three advantages originally (such as cities or urban agglomeration region) will continue to maintain and strengthen these advantages by immigration, while other areas, such as rural areas, have to face the problems of relatively smaller economy, a lower population density and an aging population society.

As for soil nutrient and land use, some studies believe that it is beneficial to the agricultural population growth, but regions with large agricultural production potential may also be experiencing population loss [53]. Areas with favorable agricultural conditions (represented by soil nutrients), carrying more agricultural population, have become the main source of out-migration for better job opportunities. Especially the inland areas with relatively backward economic development.

## Conclusions

The overall results suggest that migration in China occurred from inland to coastal areas and from rural areas to urban areas. Immigrant concentration is in important coastal areas such as the Pearl River delta and the Yangtze River delta, followed by the provincial inland capitals. In short, these regions, along with the Beijing-Tianjin area, is where the Chinese population concentrated. Regarding out-migration, this paper shows that the main spatial clustering expanded from inland areas of Guangdong, Guangxi and some of economically backward areas in the Yangtze River in 2000 to the whole provinces along the Yangtze River basin in 2010. The scale and configuration of out-migration moved from south China to central China and expanded further.

We defined 15 potential variables that affect internal migration from five aspects: economy, demography, environment, traffic and location, and land use. We concluded that the most important pull forces of migration in our study are economy scale (GDP), population density, and proportion of labour force. In the southern coastal provinces, such as Guangdong and Fujian, the roles of GDP and labour force proportion are particularly obvious. In the central provinces, the migration is most affected by population concentration level (population density).

Education is negatively correlated with internal migration, but in eastern provinces the areas with better-educated level attract more people to move in, while in the western provinces, highly educated areas face the problem of out-migration. Education became a push force for migration in the west and a pull force in the east, leading to educational gaps between regions that may continue to widen because of internal migration.

The distances to nearest provincial capital and prefecture-level city have the opposite effect on internal migration, with the former promoting net immigration and the latter net out-

migration. This is evident in the eastern coastal provinces. The population growth of the provincial capitals had spill over effects on the surrounding areas, especially in the eastern coastal areas, which would be conducive to the development of urban agglomerations. However, the immigration of prefecture-level cities is often accompanied by the out-migration of the surrounding areas, creating a competitive situation. The influence of location conditions on population migration is more obvious in the eastern coastal provinces, but the correlation is less in the western inland regions. For the eastern coastal areas, the distance from the provincial capital or prefecture-level cities is more sensitive to the migration intention.

As for the relationship between land use and internal migration, people tend to leave cropland areas and migrate to areas with higher forest cover. The trend of out-migration from cropland areas is more obvious in the southwestern inland areas, while the trend of immigration to areas with higher forest cover is more obvious in the eastern coastal provinces.

## Policy recommendations

With the migration, especially the labour force, to the densely populated urban areas with economies of scale, the population pressure in the vast rural areas will be alleviated, as young adults seek new development in urban areas, the rural population will become older. On the macro level, the government needs to guide the migration of rural population to cities in a planned manner. Specifically, for mega-cities represented by provincial capitals and above, which have spillover effects on immigration to surrounding areas, the government needs to actively promote the construction of independent comprehensive node cities in suburban areas, accelerate the integrated development of urban clusters, and reasonably control the population density of central cities. For the central cities, represented by prefecture-level cities with the tendency of absorption of immigrants from the surrounding areas, it is necessary to break the urban-rural dualism, realize urban-rural integration, promote the new urbanization and rural revitalization in concert, and promote the integrated development of urban and rural areas.

The economic development of the western region lags behind that of the eastern coastal region. The well-educated people in the western region may be motivated to migrate to the eastern region for higher salaries. In the end, the educational gap between the east and the west will continue to widen. On the macro level, the government needs to rationally guide the redistribution of population in eastern, central and western China. On the one hand, the eastern coastal regions have the advantage of talent concentration, especially the Yangtze River Delta, Pearl River Delta and Beijing-Tianjin regions. The eastern coastal regions can adjust the relevant settlement policies (Huko policies) to create high-quality talent centers and innovation highlands. The central and western inland regions should also do their best to "retain local talents". In view of the potential of immigration in densely populated areas of central and western China, several urban clusters of central and western China should be developed to absorb the local migration in central and western China. At the same time, the governments can consider the placement of densely populated industries with scale advantages in the central region and develop industries with local characteristics.

Large-scale urban-rural population migration in China can release the pressure of human activities on natural ecosystems in rural areas and promote the increase of forest cover and biomass. The "peasant-to-citizen" approach reduces the dependence of rural populations on arable land, and this study shows that populations, especially in the east, tend to migrate to areas with high forest cover. Therefore, in terms of policy, a large amount of low-yielding sloping arable land in rural areas can be considered for conversion into forestland. By optimizing the spatial development pattern of the country, green mountains and clear water are equal to mountains of gold and silver.

In any case, the active or passive migration of the Chinese rural population to the city is the general trend. Urbanization in the central and western regions can be combined with poverty alleviation policies (Liu et al. 2017) [22] and mainly guided by the government, while the eastern regions can achieve reasonable population flow by relying on the market mechanism. In short, population migration in China will continue to take place on a large scale in the future. China has a vast territory and different national conditions. In the rational planning of population mobility, we must make the best use of the situation to promote the balanced development of the population.

## Limitation of the study

As for the limitations of this study, firstly, this study adopted a stepwise regression method to exclude factors that were less correlated with internal migration and had obvious covariance between factors, which may suggest a lacked a mechanism analysis. Secondly, this study focused on the spatial heterogeneity of potential factors influencing internal migration, but the distribution characteristics and the degree of influence of such spatial heterogeneity were yet to be studied in comparison with relevant empirical studies. Finally, this study did not analyse migration resistance factors, such as migration intention.

Regarding internal migration data, in fact, there are changes after 2010. However, the authoritative source of data on Chinese migration at the counties level is mainly the decennial Chinese census. In other words, the current seventh census data (2020) are still not available. In addition, the reliability and accessibility of county-level migration data from other sources and for other years is highly problematic. Nonetheless, the overall trend of migration from rural to urban areas and from inland to coastal areas in China has not changed fundamentally. Therefore, the relevant conclusions of this paper are still valid.

## Author Contributions

**Conceptualization:** Haibin Xia.

**Formal analysis:** Emerson Augusto Baptista.

**Investigation:** Liu Qingchun, Emerson Augusto Baptista.

**Methodology:** Haibin Xia, Emerson Augusto Baptista.

**Resources:** Liu Qingchun.

**Validation:** Emerson Augusto Baptista.

**Visualization:** Emerson Augusto Baptista.

**Writing – original draft:** Haibin Xia, Liu Qingchun, Emerson Augusto Baptista.

**Writing – review & editing:** Emerson Augusto Baptista.

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
