## [Decision Letter · Decision Letter 0]

5 Aug 2022

PONE-D-22-13512Spatial heterogeneity of internal migration in China: the role of economic, social and environmental characteristicsPLOS ONE

Dear Dr. Baptista,

Thank you for submitting your manuscript to PLOS ONE. After careful consideration, we feel that it has merit but does not fully meet PLOS ONE’s publication criteria as it currently stands. Therefore, we invite you to submit a revised version of the manuscript that addresses the points raised during the review process.

We look forward to receiving your revised manuscript.

Kind regards,

Xiao-Dong Yang

Academic Editor

PLOS ONE

Journal Requirements:

"This work was supported by the project of the National Natural Science Foundation of China (grant numbers 41401657 and 41401163)"

"Specify the role(s) played

the National Social Science Fund of China(20BRK022)

Sponsored by Natural Science Foundation of Shanghai（19ZR1415200）"

4. We note that Figure 1 and 2 in your submission contain copyrighted images. All PLOS content is published under the Creative Commons Attribution License (CC BY 4.0), which means that the manuscript, images, and Supporting Information files will be freely available online, and any third party is permitted to access, download, copy, distribute, and use these materials in any way, even commercially, with proper attribution. For more information, see our copyright guidelines: http://journals.plos.org/plosone/s/licenses-and-copyright.

1. You may seek permission from the original copyright holder of Figure 1 and 2 to publish the content specifically under the CC BY 4.0 license. 

Additional Editor Comments (if provided):

I'm really sorry that your paper has gone through the review process for a long time, because there is no suitable reviewers. At present, the last two scholars who are willing to review the paper have given their comments . I respect their judgment, and give you a minor revision. Please revise the paper in time according to their suggestions.

Reviewers' comments:

Reviewer's Responses to Questions

**Comments to the Author**

1. Is the manuscript technically sound, and do the data support the conclusions?

Reviewer #1: Yes

Reviewer #2: Yes

2. Has the statistical analysis been performed appropriately and rigorously? 

Reviewer #1: Yes

Reviewer #2: Yes

3. Have the authors made all data underlying the findings in their manuscript fully available?

Reviewer #1: Yes

Reviewer #2: Yes

4. Is the manuscript presented in an intelligible fashion and written in standard English?

Reviewer #1: Yes

Reviewer #2: Yes

5. Review Comments to the Author

Reviewer #1: The authors researched and discussed a very interesting, important and topical issue.

Therefore, the question should be asked whether the authors have studied what are the main factors on internal migration of Eastern and west, for example, for work or family life.

Section Methods lacks a quantitative analysis of the use of different modes to better understand

temporal and spatial changes of the Chinese migration. Also not stated is what criterion indicates this:OLS and GWR,etc.

In the POLICY RECOMMENDATIONS, it is possible to prioritize in the two perspectives discussed - macro and micro - e.g. which activities are most important, e.g..population migration on a large scale in Urban &Rural, eastern &west .

Reviewer #2: This study calculated the internal migration, analyzed its spatial characteristics, and discussed the influencing factors of economy, demography, environment, location and transportation, land use, etc. This study is interesting and novel, and has great significance for the population management, industrial layout and population management for the policy-makers in China. I recommend publishing after minor revisions.

1. The title of Figures should put below the Figures.

2. In the introduction part, the shortcomings of previous studies and the meaning and this paper should be further summary to highlight the novelty.

3. Why the author just use the data of 2000-2010. the population migration may changed greatly after 2010. I encourage the author to explain this point.

4. I encourage the author to explain the applicability of the methods used in this study.

5. The first sentence of the conclusion part should put in introduction part. And the conclusion part should be simplified to highlight the main findings.

6. The cited reference in conclusion part should be deleted. I encourage the author to add a discussion part to compare the results with other researches, and some part of conclusion and the cited references could be changed to discussion part. moreover, the reason for the different influence of factors in different regions should be further discussed and explained.

7. the limitation of this study, I think, should be discussed.

6. PLOS authors have the option to publish the peer review history of their article (what does this mean?). If published, this will include your full peer review and any attached files.

Reviewer #1: No

Reviewer #2: No

---

## [Author Response · Author response to Decision Letter 0]

5 Oct 2022

Responses to all reviewers' comments and suggestions are in the attached document (Response to Reviewers).

---

## [Decision Letter · Decision Letter 1]

18 Oct 2022

Spatial heterogeneity of internal migration in China: the role of economic, social and environmental characteristics

PONE-D-22-13512R1

Dear Dr. Baptista,

We’re pleased to inform you that your manuscript has been judged scientifically suitable for publication and will be formally accepted for publication once it meets all outstanding technical requirements.

Kind regards,

Xiao-Dong Yang

Academic Editor

PLOS ONE

Additional Editor Comments (optional):

Reviewers' comments:

Reviewer's Responses to Questions

**Comments to the Author**

1. If the authors have adequately addressed your comments raised in a previous round of review and you feel that this manuscript is now acceptable for publication, you may indicate that here to bypass the “Comments to the Author” section, enter your conflict of interest statement in the “Confidential to Editor” section, and submit your "Accept" recommendation.

Reviewer #1: (No Response)

Reviewer #2: All comments have been addressed

2. Is the manuscript technically sound, and do the data support the conclusions?

Reviewer #1: Yes

Reviewer #2: Yes

3. Has the statistical analysis been performed appropriately and rigorously? 

Reviewer #1: Yes

Reviewer #2: Yes

4. Have the authors made all data underlying the findings in their manuscript fully available?

Reviewer #1: Yes

Reviewer #2: Yes

5. Is the manuscript presented in an intelligible fashion and written in standard English?

Reviewer #1: Yes

Reviewer #2: Yes

6. Review Comments to the Author

Reviewer #1: (No Response)

Reviewer #2: The authors have revised the manuscript carefully accordong to the comments. I recommend publishing.

7. PLOS authors have the option to publish the peer review history of their article (what does this mean?). If published, this will include your full peer review and any attached files.

Reviewer #1: No

Reviewer #2: No

---

## [Editor Report · Acceptance letter]

20 Oct 2022

PONE-D-22-13512R1 

Spatial heterogeneity of internal migration in China: the role of economic, social and environmental characteristics 

Dear Dr. Baptista:

I'm pleased to inform you that your manuscript has been deemed suitable for publication in PLOS ONE. Congratulations! Your manuscript is now with our production department. 

Kind regards, 

on behalf of

Dr. Xiao-Dong Yang 

Academic Editor

PLOS ONE